# Head and Neck Cancers Are Not Alike When Tarred with the Same Brush: An Epigenetic Perspective from the Cancerization Field to Prognosis

**DOI:** 10.3390/cancers13225630

**Published:** 2021-11-11

**Authors:** Diego Camuzi, Tatiana de Almeida Simão, Fernando Dias, Luis Felipe Ribeiro Pinto, Sheila Coelho Soares-Lima

**Affiliations:** 1Molecular Carcinogenesis Program, Brazilian National Cancer Institute, Rua André Cavalcanti, 37–6 andar, Bairro de Fátima, Rio de Janeiro CEP 20231-050, Brazil; camuzi.diego@gmail.com (D.C.); lfrpinto@inca.gov.br (L.F.R.P.); 2Departamento de Bioquímica, Instituto de Biologia Roberto Alcantara Gomes, Universidade do Estado do Rio de Janeiro, Rio de Janeiro CEP 20551-013, Brazil; tatiana.simao@uerj.br; 3Seção de Cirurgia de Cabeça e Pescoço, Instituto Nacional de Câncer—INCA, Praça da Cruz Vermelha, Rio de Janeiro CEP 20230-130, Brazil; fdias@inca.gov.br

**Keywords:** head and neck cancer, DNA methylation, microRNA, field cancerization, biomarker, tobacco, alcohol, HPV

## Abstract

**Simple Summary:**

Squamous cell carcinomas affect different head and neck subsites and, although these tumors arise from the same epithelial lining and share risk factors, they differ in terms of clinical behavior and molecular carcinogenesis mechanisms. Differences between HPV-negative and HPV-positive tumors are those most frequently explored, but further data suggest that the molecular heterogeneity observed among head and neck subsites may go beyond HPV infection. In this review, we explore how alterations of DNA methylation and microRNA expression contribute to head and neck squamous cell carcinoma (HNSCC) development and progression. The association of these epigenetic alterations with risk factor exposure, early carcinogenesis steps, transformation risk, and prognosis are described. Finally, we discuss the potential application of the use of epigenetic biomarkers in HNSCC.

**Abstract:**

Head and neck squamous cell carcinomas (HNSCC) are among the ten most frequent types of cancer worldwide and, despite all efforts, are still diagnosed at late stages and show poor overall survival. Furthermore, HNSCC patients often experience relapses and the development of second primary tumors, as a consequence of the field cancerization process. Therefore, a better comprehension of the molecular mechanisms involved in HNSCC development and progression may enable diagnosis anticipation and provide valuable tools for prediction of prognosis and response to therapy. However, the different biological behavior of these tumors depending on the affected anatomical site and risk factor exposure, as well as the high genetic heterogeneity observed in HNSCC are major obstacles in this pursue. In this context, epigenetic alterations have been shown to be common in HNSCC, to discriminate the tumor anatomical subsites, to be responsive to risk factor exposure, and show promising results in biomarker development. Based on this, this review brings together the current knowledge on alterations of DNA methylation and microRNA expression in HNSCC natural history, focusing on how they contribute to each step of the process and on their applicability as biomarkers of exposure, HNSCC development, progression, and response to therapy.

## 1. Introduction

Head and neck squamous cell carcinomas (HNSCC) are among the ten most frequent and lethal cancers among men worldwide, affecting both developed and developing countries [1]. It is usually associated with lifelong exposure to tobacco and alcohol [2,3,4], but more recently, human papillomavirus (HPV) infection has emerged as a third risk factor [4]. HPV drives oropharyngeal squamous cell carcinomas (OPSCC) with better treatment response and overall survival that led to the establishment of specific guidelines in the last TNM Classification of Malignant Tumors [5]. Nonetheless, for most HNSCC cases, the prognosis is usually poor, with a high incidence of recurrences and second primary tumors (SPT) [4,6]. Therefore, identifying the molecular mechanisms of HNSCC development and progression is of utmost relevance.

HNSCC natural history might differ according to different aspects, such as affected subsite, the treatment offered and patients’ habits. In the 1950s, Slaughter and colleagues evaluated the origin and spreading of tobacco- and alcohol-associated oral squamous cell carcinoma (OSCC), the most frequent and most accessible HNSCC [7]. In this groundbreaking work, they showed that (i) horizontal is more common than vertical spread; (ii) the surrounding benign epithelium was usually microscopically abnormal; (iii) when tumors were ≤1 cm, other in situ cancer foci or isolated islands of invasive carcinoma were found; and (iv) 11% of the cases presented synchronous macroscopic SPT in the mucosa of the upper alimentary and respiratory tracts. These observations led to the proposal of a field of cancerization, in which carcinogens precondition the epithelial lining of these tracts to transformation.

Indeed, tobacco and alcohol-associated HNSCC arise after a lifelong exposure to these risk factors. Although their full onset may take tens of years, the entire process is not completely silent. Precursor lesions can be detected and might be associated with progression risk. For decades, white (leukoplakias) and red (erythroplakias) patches in head and neck linings have been studied, and up to now, it is still difficult to estimate how often they will progress to malignant tumors and the associated factors. Initially, these lesions were evaluated exclusively according to microscopic morphological aspects. In a study from 1987, for example, the authors showed that among 42 patients with white or red patches (or both) at the vocal cords, all presented morphological alterations, ranging from keratosis to microinvasive carcinoma [8]. Among those with keratosis, keratosis with atypia, or atypia, 43%, 42%, and 80%, progressed to carcinoma in situ or microinvasive carcinoma, respectively. The time between the diagnosis of leukoplakias and/or erythroplakias and progression to malignancy also varied between 1 and 30 months. Other relevant observations from this study were the high rate of recurrence of morphological alterations (54.5%), as well as a slightly higher risk of recurrence among patients who did not stop smoking (33.3% versus 26.6% among patients who quit smoking). In a recent meta-analysis, the prevalence of oral potentially malignant disorders was estimated to be 4.47% (95% CI = 2.43–7.08), and varied between populations, with North America showing the lowest prevalence (0.11%) and Asia the highest (10.54%) [9]. The rates of malignant transformation also vary between populations and studies (ranging from 3.2% to 50%), and the strongly associated factors include lesion size, texture, color, tumor subsite, and presence of dysplasia [10]. Nevertheless, besides morphological appearance and clinical aspects, molecular biomarkers might also be used to diagnose and estimate the risk of progression, as addressed in the following sections.

The frequency of HPV-associated HNSCC (HPV-HNSCC) varies worldwide, probably because of the also variable incidence of tobacco and alcohol-associated HNSCC [11]. However, it is undeniable that HPV-HNSCC incidence rates increased in the last years, and it preferably affects younger patients, making the disease a public health issue [12,13]. The most common affected subsites include lingual and palatine tonsils. In contrast to tobacco and alcohol-associated HNSCC, HPV-HNSCC is not associated with dysplasia, lacks keratinization, and shows basaloid morphology [14,15,16]. The discontinuous basement membrane of the reticulated epithelium in tonsils enables HPV assessment to basal keratinocytes in the absence of microabrasions [17], explaining the subsite preference. When these cells replicate, HPV is passed to daughter cells, and through its transcriptome program, it disturbs the differentiation process that would occur in normal conditions [17]. E6 and E7 viral oncoproteins are crucial to high-risk HPV-induced transformation by binding to p53 and Rb, respectively, facilitating their proteasomal degradation [18,19]. When HPV DNA is integrated into the host cell genome, higher levels of viral transcripts are observed [20]. Rb degradation promoted by E7 leads to E2F release, which activates *p16INK4a* transcription [21]. Although p16 immunostaining is largely applied as a highly sensitive HPV-HNSCC surrogate marker, it is moderately specific [22]. Therefore, novel biomarkers could help discriminate between the two tumor entities.

HNSCC is highly heterogeneous regarding molecular alterations. Although differences according to tumor site and associated risk factors exist, these only partially explain this heterogeneity. In this context, epigenetic alterations have gained attention due to their frequency and site-specific pattern [23,24,25,26], being promising biomarkers. In the following sections, we bring together the current knowledge on alterations of DNA methylation and microRNA expression in HNSCC natural history, focusing on how they contribute to each step of the process and on their applicability as biomarkers of exposure, HNSCC development, progression, and response to therapy.

## 2. Epigenetic Mechanisms

Epigenetics refers to molecular alterations that do not alter the nucleotide sequence but affect gene expression. These molecular mechanisms are generally stable, reversible, and are inherited during cell division [27]. DNA methylation, noncoding RNAs, and chromatin remodeling are considered epigenetic mechanisms of gene expression control, and their dysregulation is recurrently reported in cancer [28,29]. Since epigenetic marks are overly sensitive to environmental stimuli, intrinsically associated with cell phenotype, tissue-specific, and can be detected in surrogate samples, the development of epigenetic biomarkers in oncology is promising. Much progress has been made in evaluating alterations of DNA methylation and noncoding RNAs in HNSCC. Due to analytical advantages compared with chromatin remodeling, these epigenetic marks are the focus of our discussion here.

DNA methylation is probably the most extensively studied epigenetic mechanism in cancer (Figure 1). It involves the transfer of a methyl group from S-adenosyl methionine (SAM) to the carbon 5 of cytosines followed by guanines (CpG dinucleotides), a reaction catalyzed by DNA-methyltransferases (DNMTs) [30,31]. CpG islands refer to genomic regions with an observed frequency of CpG nucleotides higher than expected, and are mainly located in promoter regions [32]. Although DNA methylation may affect gene expression, both inside and outside CpG islands [33], most studies focus on the analysis of these genomic regions. The impact of this epigenetic mark on gene expression involves the alteration of transcription factor affinity to their consensus sequence in target genes, as well as the recruitment of other proteins, such as methyl-binding domain proteins and chromatin remodelers [32]. When DNA methylation affects promoter regions, the consequence is usually gene silencing, and it is not uncommon to observe their association with heterochromatin [34]. It is also an important mechanism to maintain genomic stability through the shutdown of transposable elements. More recently, due to the widespread use of genome-wide technologies to study DNA methylation, gene bodies, enhancers, and insulators have begun to be addressed and may unveil further interconnections between DNA methylation and genome usage [35].

Among noncoding RNAs, the effects of microRNAs (miRNAs) on gene expression control are probably the most well understood (Figure 1). These molecules are about 22 nucleotides in length and recognize their target mRNAs by complementarity, usually binding to the 3′ untranslated region (UTR) [36,37,38]. MiRNA–mRNA pairing might be complete or incomplete, and the consequences are mRNA degradation and translation inhibition, respectively [37,38]. A single miRNA might have dozens of targets [38], making the impact of their dysregulation on cell phenotype quite dramatic and difficult to predict. Nonetheless, due to their tissue-specific expression pattern, frequently altered expression in cancer, and high stability in biological fluids, miRNAs are promising biomarkers in oncology.

## 3. HNSCC Etiology and Epigenetics

All etiological factors associated with HNSCC development have been shown to affect epigenetic mechanisms in different tissues. For example, both smoking and alcohol consumption were associated with global hypomethylation in HNSCC [39], while HPV-positive tumors show higher global methylation levels relative to HPV-negative tumors [40]. It is intriguing that while the expression of microRNAs can be regulated by DNA methylation [41], DNA methylation machinery can also be the target of microRNAs [42,43], creating a negative feedback. Additionally, microRNA expression can be influenced by genetic factors such as single nucleotide polymorphisms and mutations [44,45]. However, given the stochasticity of HNSCC carcinogenesis, the order of events is still far from being understood.

Furthermore, specific microRNA expression and DNA methylation signatures for tobacco [46], alcohol [47,48], and HPV infection [49,50] were established and represent not only a source of exposure biomarkers, but may also add information on carcinogenic mechanisms (Figure 2).

### 3.1. Tobacco

Tobacco components have been shown to induce epigenetic alterations (Figure 2 and Table 1). Although the molecular mechanisms are not completely elucidated, some clues have been gathered. In HeLa cells, benzo(a)pyrene (BaP) was shown to induce the deposition of activating histone marks in the promoter of long interspersed element 1 (LINE1) [51]. This was followed by the inhibition of DNMT1 binding to these genomic regions, and by the proteasome-mediated degradation of this enzyme. As a result, BaP induced LINE1 hypomethylation [51]. By binding to nicotinic cholinergic receptors, nicotine was shown to decrease *Dnmt1* mRNA and protein levels and induce demethylation of specific loci [52]. Other tobacco components may induce DNA hypermethylation. The tobacco-specific nitrosamine 4-(methylnitro-samino)-1-(3-pyridyl)-1-butanone (NNK) can induce AKT activation that phosphorylates and inhibits GSK3β [53]. Once inactivated, GSK3β can no longer phosphorylate DNMT1, which is not marked for proteasomal degradation, and it is therefore stabilized. Another mechanism by which tobacco components may induce hypermethylation is via DNA damage; Dnmt1 was shown to be recruited to DNA repair sites to restore epigenetic marks [54]. In vitro experiments in HeLa and mouse embryonic stem cells corroborated these findings, showing that following double-strand breaks, half of the strands repaired by homologous recombination become hypermethylated, which is mediated by DNMT1 [55]. Cigarette smoke condensate can also induce Sp1 expression and DNA binding [56]. Since Sp1 is a transcription factor that recognizes CG-rich genomic regions [57], its binding may prevent DNA methylation in these regions [58,59]. Finally, tissue hypoxia induced by cigarette smoke carbon monoxide can induce global DNA hypomethylation. Hypoxia-inducible factor 1α (HIF-1α) induces the expression of methionine adenosyltransferase 2A (MAT2A, involved in SAM synthesis), and as a consequence, SAM levels are decreased, impairing DNA methylation reactions [59,60].

Epigenome-wide association studies (EWAS) on different tissue types were carried out to identify tobacco epigenetic signatures. Using blood samples and the HumanMethylation27 assay, Breitling and colleagues identified one CpG locus annotated to *F2RL3* that was hypomethylated in heavy smokers relative to never smokers. Former smokers showed methylation patterns similar to never smokers [140]. A following study applying the HumanMethylation450 beadchip, both in lymphoblasts and alveolar macrophages, identified one differentially methylated probe in *AHRR* gene body when comparing smokers and nonsmokers [141]. Similar findings for *AHRR* methylation and additional differentially methylated CpGs in *CYP1A1* and *GFI1* were observed in newborns following maternal smoking [142]. Additionally, in blood-derived DNA analyzed by HumanMethylation 450 assay, a meta-analysis with 15,907 individuals from 16 cohorts identified 18,760 differentially methylated CpGs between current and never smokers, including CpGs mapped to previously identified genes, such as *AHRR*, *RARA*, *F2RL3*, and *LRRN3* [143]. A total of 2568 CpGs were also differentially methylated between never and former smokers. The consistent association between the methylation profile of genes involved in xenobiotic metabolism and smoking adds a biological plausibility to the results obtained.

An EWAS carried out in buccal cells from 790 women showed that smoking also leaves a DNA methylation signature in epithelial cells, which can be even more prominent throughout the genome than that determined in blood cells [144]. In this study, the top-ranked CpG sites showed a hypomethylation profile in smokers and affected genes such as *AHHR*, *F2RL3*, and *CYP1A1*, previously identified in blood-based EWAS/smoking studies. While hypomethylated genes were not enriched for specific cell pathways, hypermethylated CpGs were enriched for genes bivalently marked in human embryonic stem cells, and for binding sites of transcription factors associated with stem cell differentiation. These smoking-responsive CpG sites were also differentially methylated in tobacco-associated cancers, suggesting a contribution of these alterations to tumor development.

In OPSCC patients, the blood of 106 never and 303 ever smokers were compared and revealed the differential methylation of 52 CpG sites [145]. Of these, 49 sites had lower methylation levels in smokers, including a CpG mapped to *AHRR*. A similar analysis was performed to predict survival, and 18 CpG sites annotated to three genes (*GFI1*, *SPEG*, and *PPT2*) were differentially methylated according to both survival and smoking history. Additionally, in OPSCC patients, a DNA methylation score for smoking based on five EWAS was applied on 364 blood samples and was able to explain 51.1% of phenotypic variance [146]. This score showed an accuracy of 0.70 to predict mortality, slightly improving the accuracy of smoking self-report (0.67).

In gene-specific studies, *p16INK4a* hypermethylation was already reported in 9.7% of oral mucosa and/or tongue of healthy smokers [147]. Similarly, higher methylation levels of *p16INK4a* and *MYOD1* were observed in HNSCC from smokers relative to nonsmokers [61]. *SFRP4* hypermethylation seems to be more common in never and former smokers than in current smokers, and it is also independently associated with HPV16 infection [62]. In a study evaluating the methylation profile of 10 genes (*p16INK4a*, *DAPK*, *GSTP1*, *RASSF1*, *BRCA1*, *ECAD*, *MLH1*, *MINT1*, *MINT2*, and *MINT31*), both the methylation index (determined by the frequency of hypermethylated promoters) and the frequency of CpG island methylator phenotype-high (CIMP-high, at least five hypermethylated genes) cases were higher among smokers relative to nonsmokers [78]. In opposition to the hypomethylation profile observed in the EWAS, *CYP1A1* hypermethylation was more frequently observed in HNSCC from smokers relative to healthy tissues from controls [63]. When promoter methylation and gene deletion of *RBSP3*, *LIMD1*, and *CDC25A* were analyzed together in HNSCC, the frequency of alterations was higher in smokers than in nonsmokers [64].

MicroRNAs are also targets of dysregulation by different tobacco components, varying according to route of exposure and head and neck subsite. Immortalized human oral keratinocytes OKF6/TERT exposed to cigarette smoke condensate or chewing tobacco extract showed six and ten differentially expressed microRNAs relative to nontreated cells, respectively, with none in common between treatments [148]. After treatment with smoke condensate, members of the miR-200 family were downregulated. These data are consistent with their downregulation observed in OSCC patients with smoking/chewing tobacco habits [65]. The exposure of oral keratinocytes OK4 to cigarette extract further showed the upregulation of miR-101, miR-181b, miR-486, and miR-1301 [149].

The exposure of HNSCC cells to NNK induced the expression of miR-21 and miR-155, while decreased miR-422a expression. Overexpression of miR-155 was also associated with the habit of chewing tobacco [66]. MiR-21 expression was further associated with *MSH2* silencing [150]. MSH2 is a member of the mismatch repair pathway, and its downregulation was associated with poor survival of HNSCC patients [151]. NNK can also induce the expression of miR-944 that targets *CISH*, a suppressor of the STAT pathway [152,153]. *CISH* inhibition by miR-944 induces STAT3 activation, modulating the inflammatory response and increasing tumor malignancy [153]. MiR-30a and miR-379 expression levels were downregulated in OSCC cells exposed to NNK. Both miRNAs are downregulated in OSCC relative to adjacent nontumor tissue and target *DNMT3B*. This can lead to the hypermethylation of *ADHFE1* and *ALDH1A2* promoters, and their consequent downregulation [154]. Since both genes are involved in alcohol metabolism, these data suggest an interesting potential mechanism for the synergy between alcohol and tobacco in HNSCC.

MiR-29c downregulation was associated with smoking and worse survival in laryngeal squamous cell carcinoma (LSCC) patients [67]. Additionally, in LSCC, smokers showed miR-202-3p upregulation and downregulation of miR- 4768-3p, miR-548aa, and miR-3713 [68]. Among HNSCC patients, miR-34 family was shown to be upregulated in tumors from smokers relative to the adjacent tissues, while no differences were observed for nonsmokers [69]. Smoking or chewing tobacco was associated with miR-429 and miR-141 downregulation in OSCC patients [148]. In OPSCC, a comparison between smokers and nonsmokers HPV-positive patients revealed 38 differentially expressed miRNAs [70]. Mir-133a-3p was among the downregulated miRNAs, and its expression was further shown to be decreased by treatment with cigarette smoke condensate and associated with EGFR and HuR expression in HNSCC cell lines [70].

### 3.2. HNSCC Etiology and Epigenetics: Alcohol

Epigenetics and metabolism are two substations in the same system, intertwined. This is probably the simplest way to understand how alcohol may induce aberrant DNA methylation (Figure 2, Table 1). Epigenetic writers and erasers are regulated, among other mechanisms, by the availability of their substrates and cofactors. Ten-eleven translocation (TET) enzymes are α-ketoglutarate-dependent oxygenases involved in active DNA demethylation [155]. Due to this cofactor dependency, TET activity is regulated by different factors that affect the tricarboxylic acid cycle (TCA), including hypoxia [156]. For DNMTs activity, the levels of the methyl donor SAM are determinant. SAM is produced by one-carbon metabolism, which relies on the dietary intake of folate and B vitamins. Ethanol is an inhibitor of folate absorption, besides inhibiting different steps of one-carbon metabolism, leading to a reduction of SAM levels. Therefore, alcohol consumption may impair the methylation reactions catalyzed by DNMTs. Indeed, global DNA hypomethylation was observed in the liver of ethanol-fed rats [157]. While the higher intake of folate has already been associated with a decreased risk of HNSCC development [158], the interaction between folate, alcohol, and DNA methylation has been neglected. Nevertheless, several studies have shown an association between DNA methylation patterns and alcohol consumption in different settings, and surprisingly, it is not restricted to DNA hypomethylation.

A recent EWAS, including 13,317 individuals, proposed a 144-CpG signature in whole blood able to discriminate current heavy alcohol drinkers [47]. This model was used to predict all-cause mortality among OPSCC patients, but although it explained 16.5% of phenotypic variance, it showed a similar predictive accuracy to self-reported data [146]. Additionally, in the blood of OPSCC patients, an EWAS identified three new CpG sites and 40 differentially methylated regions associated with alcohol consumption [145]. The gene with the lowest *p*-value for both analyses was *SLC7A11*, with the methylation levels of one of its CpG decreasing 0.10% per unit increase in alcohol consumption. This gene plays a role in glutathione synthesis, an antioxidant that can be depleted by excessive ROS production during alcohol metabolism [159].

Other gene-specific studies have suggested an association between alcohol consumption and DNA methylation profiles in HNSCC. *P15INK4b* methylation was detected in the oral and throat rinses of 48% of HNSCC patients and 68% of healthy individuals who smoked and/or drank alcohol, but in only 8% of healthy nonsmokers and nondrinker individuals [71]. *FUSSEL18* and *SEPT9* methylation were also associated with alcohol and tobacco consumption history in HNSCC [72]. In a meta-analysis including 18 studies with different sample sources (tumors, liquid biopsies, and buccal scraping), *DAPK* methylation was significantly associated with alcohol consumption (OR = 1.85, 95% CI = 1.07 ± 3.21) [73]. *SFRP1*, a member of the secreted frizzled receptor protein family, is hypermethylated in HNSCC and this event is more common in ever drinkers than in never drinkers [62]. Other genes, such as *p16INK4a*, *RASSF1A*, and *FANCF* are also more frequently hypermethylated in HNSCC associated with alcohol consumption [61,74]. On the other hand, *MLH1* methylation seems to be more common in individuals who drink less than 100 mL of alcohol/day in comparison with those who drink larger amounts [75].

Recently, alcohol consumption was associated with an aberrant DNA methylation profile in the blood of HNSCC survivors [160]. This dysregulation affected genes encoding histone and mitochondrial ribosomal proteins, as well as genes involved in toll-like receptor signaling, suggesting that alcohol intake can have a systemic effect, even after treatment.

In comparison with data showing an association between alcohol and DNA methylation in HNSCC, data for such an association with microRNA expression are still limited. As observed for DNA methylation, alcohol metabolism is also related to the expression of microRNAs. ADH1B is a key controller of alcohol metabolism, affecting acetaldehyde levels in the body [161], and its expression was shown to be regulated by miR-205-5p in HNSCC [162]. Other genes related to alcohol metabolism, such as *ADHFE1* and *ALDH1A2*, are indirectly regulated by miR-30a and miR-379-mediated *DNMT3B* downregulation in OSCC [154], as previously mentioned, revealing a crosstalk between the methylation machinery and microRNAs in the xenobiotic response. The dysregulation of genes involved in alcohol metabolism by microRNAs was also associated with aggressive phenotypes. *ADHFE1* and *ALDH1A2* higher expression, as a consequence of miR-30a and miR-379 induction, were associated with proliferation inhibition of OSCC cell lines [154]. It is important to note that ALDH1A2 does not metabolize acetaldehyde efficiently, but it is important in the metabolism of retinoic acid [163]. In addition, *ALDH1A2* is a tumor suppressor candidate in HNSCC [164].

Tumor suppressor genes may also be targets for microRNAs upregulated in HNSCC of alcoholic patients relative to nonalcoholic patients, as shown for miR-30a and miR-934. When miR-30a is inhibited, *BNIP3L*, *PRDM9*, and *SEPT7* upregulation is observed, while miR-934 inhibition leads to *HIPK2*, *HOXA4*, and *MLL3* upregulation. MiR-30a and miR-934 inhibition were also associated with increased sensitivity to cisplatin and decreased cell invasion of HNSCC cell lines [48]. Other microRNAs upregulated in HNSCC of alcoholic patients relative to nonalcoholic patients are miR-3178, miR-675, miR-101, miR-126, miR-3164, and miR-3690 [48].

Finally, miR-375, a classically downregulated microRNA in HNSCC [165], was shown to be upregulated in alcoholic patients [76]. MiR-375 targets the *AEG-1* [166] and *LDHB* oncogenes [167] and its lower expression was previously associated with invasion and worse prognosis [168].

### 3.3. HNSCC Etiology and Epigenetics: HPV

Regarding HPV infection, viral oncoproteins were shown to modulate levels of DNMTs (Figure 2) [169]. E6 knockdown in HPV16 infected cervical cancer cell lines led to decreased DNMT1 expression, which was mediated by higher p53 levels [170]. Similarly, HPV16 E7 oncoprotein was also shown to stimulate Dnmt1 activity, both by promoting E2F activity through Rb downregulation [171], and by direct binding [172], with a consequent decreased E-cadherin expression [173]. Therefore, it is expected that HPV infection affects DNA methylation patterns.

Different studies addressed genome-wide DNA methylation differences between HPV-positive and HPV-negative HNSCC (reviewed in [174]). Hypermethylation was more frequent in HPV-positive HNSCC samples and cell lines relative to HPV-negative counterparts [175,176]. Furthermore, a DNA methylation signature for HPV infection composed of five CpG sites was proposed and showed high specificity to discriminate HPV-positive and HPV-negative HNSCC in different datasets, as well as a capacity to predict survival similar to HPV status, determined by viral mRNA levels [175]. In OPSCC, another DNA methylation signature composed of differentially methylated regions in the promoters of five genes (*ALDH1A2*, *OSR2*, *GATA4*, *GRIA4*, and *IRX4*) was shown to predict overall and progression-free survival, independently of HPV status [177].

Promoter methylation of genes recurrently reported altered in HNSCC also seem to differ according to HPV infection. For *p16INK4a*, results are discordant between studies, including hypermethylation [77,78], hypomethylation [79,80], and no differences found in HPV-positive relative to HPV-negative HNSCC cases [178]. This may be a consequence of both the different sample sources (tumors and saliva) and the different tumor subsites used, including oral cavity, oropharynx, both combined, and even HNSCC generally. Therefore, further studies are needed to resolve these discrepancies. *RASSF1A*, *TIMP3*, and *PCQAP*/*MED15* were found to be hypomethylated in the saliva of HPV-positive HNSCC [79], while *RXRG*, *CTNNA2*, *GHSR*, and *ITGA4* were reported hypermethylated in HPV-positive oropharyngeal tumors [80]. Global DNA methylation also differs according to HPV status, with LINE1 methylation levels being higher in HPV-positive relative to HPV-negative tumors [179,180,181,182,183]. DNA methylation differences in transposable elements according to HPV status were recently shown to affect gene expression, which was further associated with prognosis [183].

Interestingly, DNA methylation patterns also differ when HPV DNA is integrated or episomal, with integration-positive tumors showing a profile more similar to that of HPV-negative HNSCC [20]. *BARX2* and *IRX4* were hypermethylated and silenced, while *SIM2* and *CTSE* showed hypomethylation and higher expression in episomal HPV-HNSCC. Finally, differentially methylated genes were enriched for the cAMP response element-binding protein (CREB) pathway, suggesting different carcinogenic biological mechanisms, depending on whether HPV is integrated.

DNA methylation was further associated with miRNA dysregulation in HPV-associated HNSCC [184]. MiR-139-3p was shown to be downregulated in HPV-positive tumors and cell lines relative to their HPV-negative counterparts, and it was shown to target HPV oncoproteins. In vitro experiments with miR-139-3p mimic led to E6 and E7 downregulation, reduced p16, and increased p53 protein levels. Interestingly, miR-139-3p colocalizes with *PDE2A*, whose promoter is hypermethylated in HPV-positive HNSCC. Corroborating this coregulation, treatment of HPV-positive cell lines with a demethylating agent induced miR-139-3p expression. MiR-375, previously described as a tumor suppressor whose expression is associated with alcohol intake, also seems to be suppressed by DNA methylation in HPV-mediated carcinogenesis [185]. Since HPV E6 and E7 are among miR-375 targets, downregulation of this miRNA contributes to a higher expression of these oncoproteins and a lower expression of their cellular targets, p53 and Rb [186].

Differentiation-associated miRNAs might also take part in HPV-mediated carcinogenesis [187]. As previously mentioned, HPV disturbs the differentiation of infected cells, and this was shown to be mediated, at least in part, by miR-203 downregulation. This miRNA regulates differentiation by targeting p63 and its expression can be downregulated by E7 via the blockage of MAPK pathway [187].

Although the miRNA biogenesis machinery was found to be dysregulated in cervical cancer [188], this is yet to be investigated in HPV-associated HNSCC. Nonetheless, miRNA signatures according to HPV status were reported in HNSCC. By comparing HPV-positive and HPV-negative tumors with adjacent tissue, 282 and 289 differentially expressed miRNAs were identified, respectively [189]. Although these numbers were similar, when specific survival-associated miRNAs were evaluated, the expression of 39 miRNAs was associated with prognosis among HPV-positive patients, 16 among HPV-negative patients, and no miRNA was shared. The miRNA survival signatures also differed in accuracy (0.92 for HPV-positive and 0.72 for HPV-negative) and associated signaling pathways. The poor prognosis miRNA signature for HPV-positive patients was associated with immune-related pathways. In contrast, in HPV-negative cases, it was associated with metabolism and oncogenic pathways, such as WNT and NOTCH signaling.

More recently, the direct comparison of HPV-positive and HPV-negative HNSCC revealed the differential expression of 59 miRNAs, three of which (miR-99a-3p, miR-411-5p, miR-4746-5p) were also associated with overall survival in a multivariate Cox regression analysis [81]. The Gene Set enrichment analysis performed with their potential mRNA targets showed that different pathways were likely to be dysregulated, with genes involved in epithelial–mesenchymal transition being downregulated by miR99a-3p-high/miR-411-5p-low/miR-4746-5p-high. This profile may help to explain the favorable prognosis of these patients.

Oropharyngeal tumors were also compared according to HPV status, revealing the upregulation of miR-320a, miR-222-3p, and miR-93-5p, as well as the downregulation of miR-199a-3p//miR-199b-3p, miR-143, miR-145, and miR-126a in HPV-positive tumors [82]. In tonsillar squamous cell carcinoma, 36 miRNAs were found to be differentially expressed according to HPV status [190].

When comparing HPV-positive HNSCC cell lines with HPV-negative and human foreskin keratinocytes, miR-363, miR-33, and miR-497 were upregulated while miR-155, miR-181a, miR-181b, miR-29a, miR-218, miR-221, miR-222, and miR-142-5p were downregulated [191]. This miRNA dysregulation was associated with the expression of HPV oncoproteins E6 and E7. The analysis of exosomes further revealed that miR-205-5p and miR-1972 are exclusively detected in HPV-positive and HPV-negative HNSCC cell lines, respectively [192].

Regarding minimally invasive biopsies, a miRNA panel composed of miR-9, miR-134, miR-196b, miR-210, and miR-455 evaluated in the saliva of HNSCC patients was shown to discriminate HPV-positive and HPV-negative cases [193].

Not only previously annotated miRNAs show differential expression according to HPV status. By applying a customized in silico pipeline, Rock and colleagues identified 146 novel miRNAs in samples from HNSCC patients, with three (HNnov-miR-2, HNnov-miR-30, and HNnov-miR-125) being significantly downregulated in HPV-positive tumors [194].

Although most studies bring consistent associations between risk factor exposure and aberrant epigenetic profiles in HNSCC, causal effects were not determined. In the attempt to overcome this limitation, in vivo studies should be performed with a focus on head and neck tissues. As an example, the treatment of mice with dibenzo[def,p]chrysene (DBP), a tobacco carcinogen, was able to induce DNA methylation alterations of 30 loci in oral tissues [195]. *Fgf3*, frequently amplified in HNSCC, was one of the affected genes and showed hypomethylation accompanied by induced expression. Considering that tobacco smoke contains more than 60 different carcinogens [196], alcohol may contribute to carcinogenesis by different mechanisms [197], HPV oncoproteins regulate key epigenetic enzymes [169,170,172,173], and that how all these factors interact is largely unknown, much is yet to be learned considering the effects of risk factor exposure on epigenetic mechanisms in head and neck tissues.

## 4. Epigenetic Biomarkers in HNSCC

As already mentioned, epigenetic changes coordinate the cellular response to environmental exposures, such as cancer risk factors [198,199]. Thus, even before the histological transformation or acquisition of driver mutations, we can have stochastic or targeted epigenetic changes, the latter sensitizing the cell to transformation, building the pre-cancer epigenetic signatures of the cancerization field (Figure 3) [200,201]. Several of these changes would not only favor the initial gain of a pre-cancer cell phenotype, but would also be maintained or altered after malignant transformation, tumor progression, and metastasis. Thus, epigenetic signatures are not just a consequence of risk factor exposure, or linked to the tissue of origin, but a unique product of these two factors combined [202]. In addition, unlike genetic changes, these signatures are plastic, and can vary from healthy tissue through the cancerization field and tumor progression in a qualitative or quantitative manner depending on the mechanism.

Given this scenario, it is impossible not to notice the relevance and potential of epigenetic changes as biomarkers in oncology [203,204]. In the next section, we focus on epigenetic changes as potential biomarkers in clinical use without emphasizing their contribution to gene and biological regulation. Interestingly, due to the aforementioned plastic characteristic of epigenetic marks, it is possible to identify common epigenetic changes, perhaps with different levels, aimed at more than one biomarker objective (e.g., diagnosis and prognosis). Here, we seek to bring a snapshot of epigenetic changes as biomarkers, technical limitations, and advantages, together with future application prospects.

### 4.1. Biomarkers of the Cancerization Field: From Identification to Surgical Margin and Recurrence

The molecular landscape of HNSCC carcinogenesis has been unraveled, and several candidates for biomarkers to identify pre-cancer lesions in the cancerization field are emerging (Table 1). Except for HPV-positive oropharyngeal tumors, the main genetic alterations in HNSCC are *TP53* mutations, present in up to 84% of cases [23]. *TP53* mutations have become the focus of many studies (perhaps the majority) aimed at genetic characterization of the cancerization field [205,206,207]. Although less frequently, these changes can already be found in the adjacent nontumoral dysplastic [208,209] and nondysplastic tissues of HNSCC patients [209], becoming a frequent event in key moments of the dysplasia–carcinoma transition [210].

However, *TP53* mutations can be detected throughout the entire gene without clear hotspots [211], requiring techniques that are resourceful and time-consuming, such as next-generation sequencing. In any case, we would still have a high number of false negatives due to wild-type *TP53* tumors and the HPV-positive group, requiring an even larger gene panel to cover the high genetic heterogeneity of this group [23]. By contrast, studies on esophageal squamous cell carcinoma (ESCC), which shares the same histology, epithelium of origin and concomitant exposure to the same risk factors as HNSCC, suggest not only that epigenetic alterations such as DNA methylation are more frequent, but could even precede the genetic changes in precursor lesions [212,213].

In HNSCC precursor lesions, *p16INK4a* is one of the first genes inactivated either by deletion or DNA methylation [23,214]. *P16INK4a* promoter hypermethylation was already extensively evaluated, but results vary widely, probably due to the divergence of the target location within the promoter region, and the methodologies applied, mostly methylation-specific PCR (MSP). Despite cost-effectiveness, it is a technique with a predominant qualitative profile, and may suffer a strong bias due to primer design, for example [215]. Interestingly, the methylation profile of *p16INK4a* promoter region does not seem to be associated with HPV status assessed by p16 immunohistochemistry, whereas gene body hypermethylation does [216]. However, *p16INK4A* gene body hypermethylation is likely to occur after its overexpression [217], being more suitable as an infection marker. In addition, deletion is the most frequent *p16INK4a* inactivation mechanism in HPV-negative HNSCC (57%), and practically nonexistent in HPV-positive tumors [23], in which p16 immunostaining is used as a surrogate biomarker of HPV infection [22].

Other genes with aberrant methylation in precursor lesions include *MGMT*, *p14ARF*, *p15INK4b*, *RARB*, and *SOCS-3* [83,84,86,218]; *p16INK4a* and *MGMT* hypermethylation can already be detected in oral rinse [84] and blood [85] of individuals with precursor lesions or a history of OSCC, reducing the invasiveness of the procedure. Higher methylation levels of *p16INK4a*, *p14ARF*, and *p15INK4b* were observed in LSCC precursor lesions [218]. *RARB* is hypomethylated in 27% of normal adjacent HNSCC tissues and in 53% of oral precursor lesions [86]. For more detailed reviews on the evaluation of these genes as biomarkers of the cancerization field, please refer to [219,220].

*ZNF282* and *PAX1* hypermethylation were originally proposed as biomarkers of cervical carcinoma precursor lesions [221,222,223]. In the oral cavity, *ZNF582* and *PAX1* methylation increase as the epithelium goes through transformation, being able to differentiate the visually normal mucosa or with minimal histological changes (hyperplasia or mild dysplasia) from those more advanced pre-cancer changes (moderate or severe dysplasia) and the tumor itself, with sensitivities of 85% and 72% and specificities of 68% and 86%, respectively [87]. Genes involved in the WNT and the MAPK signaling pathways show a similar profile, with aberrant methylation being observed as the precursor lesions progress [224], and may be the focus of future studies.

Although, to the best of our knowledge, whole-genome bisulfite sequencing has not yet been applied to identify DNA methylation alterations throughout HNSCC natural history, next-generation sequencing (NGS) was used to assess the methylation of specific target genes in oral brushing from healthy donors, low-grade squamous intraepithelial lesions (LG-SIL), high-grade SIL (HG-SIL), and OSCC patients [88]. Among the seven targets selected, *ZAP70* was hypermethylated in 100% of OSCC and HG-SIL cases, while *GP1BB* hypomethylation was detected in 90.9% of these individuals. Healthy donors did not show alterations of these genes, while LG-SIL showed an intermediate profile. Although the results were encouraging, this was a preliminary study with a limited number of samples.

In addition to DNA methylation, the expression of microRNAs can be used to identify the cancerization field. MiR-204, miR-31, miR-31*, miR-133a, miR-7, miR-206, and miR-1293 showed overexpression in one or more types of OSCC pre-cancer lesions relative to healthy mucosa [89]. Interestingly, miR-204 is downregulated in oral submucous fibrosis and upregulated in leukoplakia and lichen planus [89]. Similarly, representing this specific pre-cancer lesion expression signature, miR-423 and miR-34b are downregulated in lichen planus, while miR-26c and miR-29a are also downregulated in lichen planus, but overexpressed in leukoplakias [225].

Besides their potential to be applied in the early diagnosis, cancerization field biomarkers can also be used to establish surgical margins, reducing the chance of recurrences. Although histopathological evaluation is widespread, 10–30% of HNSCC patients with free surgical margins will recur [226,227,228], which is one of the main causes of mortality among HNSCC patients [228,229]. Therefore, a molecular screening of cancerization field alterations may predict this phenomenon and, therefore, improve patients’ prognosis [229]. Approximately 25% of HNSCC surgical margins contain genetic alterations [230], and *TP53* mutations were already associated with recurrence [231]. Epigenetic alterations, however, also show great potential to predict relapse.

*P16INK4A* and/or *MGMT* promoters were found hypermethylated in 32% of HNSCC surgical margins, despite all being considered negative in the histopathological analysis [232]. When evaluated together at the tumor margin, the methylation levels of *EDNRB* and *HOXA9* are predictors of locoregional recurrence-free survival, with their hypermethylation increasing the chance of recurrence approximately three-fold [125].

The analysis of a DNA methylation panel composed of three genes (*DCC*, *CCNA1*, and *p16INK4A*) in surgical margins was able to predict all recurrences in five out of 47 HNSCC patients [126]. In OSCC patients, *ZNF582* and *PAX1* methylation levels are reduced after treatment, and interestingly they rise again at the recurrence site a few months before the diagnosis [87]. In HNSCC surgical margins, the promoter methylation of another Homeobox gene, *PAX5*, increases the risk of locoregional recurrence approximately four-fold [127]. While most methylation studies focus on the promoter region of *CDKN2A* (which encodes both *p14ARF* and *p16INK4A*), its gene body hypermethylation and increased *p14ARF* expression in LSCC were associated with a lower incidence of locoregional recurrence [128]. More recently, a genome-wide DNA methylation screening identified 392 differentially methylated regions associated with recurrence when OSCC free surgical margins were analyzed. Accuracy of up to 98% was achieved to discriminate relapsed and nonrelapsed OSCC patients when a 14-CpG panel signature was applied [233].

MiRNA expression also shows recurrence predictive value. Oral precursor lesions of patients who relapsed showed miR-375 downregulation relative to those who did not relapse, presenting an accuracy of 95.7% and sensitivity of 90% to discriminate the two groups. Despite outstanding discrimination values, it is important to highlight that only six patients were included in the nonrelapsed group [129].

Although little explored in general, the oral cavity has the most well-defined and explored epigenetic cancerization field among HNSCC. This might be a consequence of the relatively easy access to the tumor site and the visual identification of precursor lesions, such as leukoplakias. In addition to the need to gather more data on other head and neck sites and to consistently validate the findings, studies on epigenetic alterations of the cancerization field should adopt more sensitive methodologies, and explore alterations in the mucosa of individuals with low and high exposure to risk factors, different types of precursor lesions, and cancer. Further, large-scale platforms may add new information and help to identify new biomarkers.

### 4.2. Diagnostic Biomarkers

In comparison to the few common genetic alterations found in HNSCC generally, DNA methylation shows more frequent and subsite-specific changes, an important asset for molecular diagnosis [24,212,213,234]. *P16INK4A* is hypermethylated in 27–44% of HNSCC [93,97,98,235]; however, it has already been reported that almost half of the samples could not be analyzed due to loss of heterozygosity [98,236]. *CDH1* is hypermethylated in 35–43% [93,95,235], *DAPK* in 43% [73], and *MGMT* in one-third of HNSCC cases [98,103,113,235,237], and hypermethylation of the latter is independent of HPV status. This hypermethylation profile is not exclusive to the tumor and can be observed in the cancerization field, as already mentioned. For example, adjacent tissues show *DAPK* hypermethylation in 20% of samples [73]. *GNAT1* and *SEMA3B* are hypermethylated in 75% and 77% of OSCC and 56% and 38% in adjacent tissues, respectively. These and other studies emphasize the importance of applying a sensitive methodology to define and differentiate this hypermethylation status [118,238]. Aberrant methylation in other genes, such as *FAM135B* and *ZNF610* hypermethylation and *HOXA9* and *DCC* hypomethylation, show accuracies of 79%, 93%, 93%, and 93% in distinguishing adjacent nontumor tissues and HNSCC, respectively [239]. Table 1 brings additional candidate diagnostic biomarkers based on aberrant DNA methylation in HNSCC.

MicroRNA dysregulation was also evaluated as a diagnostic biomarker. In LSCC, miR-375 is downregulated, while miR-21 is overexpressed, and their expression ratio has a 99% accuracy to discriminate tumor and adjacent nontumor tissue [122]. In OSCC, miR-204 and mir-144 are overexpressed, while miR-193b-5p and miR-370-3p are downregulated, and have an individual accuracy of 91%, 85%, 57%, and 61%, respectively, and a combined accuracy of 92% to discriminate the tissues [123]. Combining the expression of miR-657 and miR-1287, an accuracy of 97% and sensitivity of 86% were achieved in early stage LSCC [124].

Despite the promising discrimination values mentioned above, few studies share the same findings on miRNA dysregulation in HNSCC (see Appendix A and Figure 4) [240]. This underscores the high intertumoral heterogeneity and the need for establishing subsite-specific biomarkers.

DNA methylation and microRNAs expression can be sensitively analyzed in fluids, decreasing the diagnosis’s invasiveness and even allowing screening programs in groups at risk. Perhaps the greatest group at risk for HNSCC development is composed of individuals with a previous HNSCC diagnosis, since 9–16% develops a SPT, and 49% in another region of the head and neck [255,256,257]. Noninvasive diagnosis can facilitate the follow-up and SPT early detection, increasing the chances of survival.

With the exception of the oral cavity, access to other sites of the head and neck might be challenging, and less invasive diagnostic techniques are required. The assessment of epigenetic biomarkers in body fluids, such as saliva and blood, has emerged as an alternative and may even impact the biomarker performance [258]. HNSCC patients already show a global hypomethylation in whole blood, estimated by the evaluation of LINE1 transposable elements. However, in addition to being a generic tumor biomarker [259,260,261,262,263,264], global methylation levels showed overlap between HNSCC patients and controls, jeopardizing their discrimination capacity [110]. *THF* methylation profile was able to differentiate with 93% accuracy (86% sensitivity) individuals without cancer from those with OSCC or OPSCC using oral rinse [265]. The methylation profile of *MED15* promoter region in the saliva of HNSCC patients showed an accuracy of 63–70% [266]. In the plasma of LSCC patients and individuals without cancer, 26 differentially expressed microRNAs were identified, 17 overexpressed, and 9 downregulated in patients [243].

In addition to tumor diagnosis, the DNA methylation profile can also help to identify the risk factors associated with tumor development. By evaluating the saliva of HPV-negative HNSCC patients, *RASSF1A*, *p16INK4A*, *TIMP3*, and *PCQAP*/*MED15* genes were found hypermethylated relative to healthy individuals and showed a sensitivity of 71% to discriminate the groups [79]. The same genes are hypomethylated in the saliva of HPV-positive HNSCC patients relative to healthy individuals and distinguished the groups with a sensitivity of 80% [79]. A panel with the methylation profile of *p16INK4A*, *RASSF1A*, *TIMP3*, and *PCQAP*/*MED15* genes in saliva was able to differentiate OSCC and OPSCC patients from individuals without cancer with 91% and 92% sensitivity and 99% and 92% specificity, respectively [117].

A common feature of HNSCC is the marked disparity in the overall survival of patients diagnosed in early stages compared to the more advanced ones. Screenings widely applied in oncology, such as those for breast [267] and cervical [267] cancer early diagnosis have sensitivities of approximately 96%. Studies aimed at tumor screening in the oral cavity showed that despite a good number of pre-cancer lesions were identified (~10%), the number of carcinomas found was low [268], with a sensitivity of 71–76% [269,270]. However, even with the low frequency of tumors, reduced staging and improved overall survival were observed in patients due to screening, showing that it can be worthwhile [269].

### 4.3. Biomarkers of Treatment Response and Prognosis

Despite advances in oncology, the average five-year survival of HNSCC patients remains the same, 13–57%, varying according to the affected primary site [271,272]. The therapeutic approach depends on the location, resectability, and strategies focused on organ preservation, including surgery, chemotherapy, and radiotherapy [273]. The gold-standard treatment for the disease is surgery, which may be combined with radiotherapy and, in locally advanced cases, chemotherapy based on taxanes and platinum [273].

In HNSCC, molecular changes and etiology can drive treatment choice. Patients with HPV-positive OPSCC respond better to therapy, and treatment deintensification was already proposed [274]. By contrast, smokers and drinkers generally show a poor response to treatment [275,276,277]. In this context, epigenetic changes have been proposed as surrogate markers of therapeutic response and epigenetic drugs were shown to resensitize cell lines to conventional therapy.

DNA methylation alterations have been recently shown to predict locoregional recurrence after surgery, for example. By evaluating the methylation levels of 13 genes by bisulfite NGS in pre-operative OSCC samples collected by oral bushing, Gissi and colleagues showed that five CpG sites in three genes (*EPHX3*, *ITGA4*, and *MiR193A*) were able to predict adverse outcomes with high accuracy (AUC = 0.85) [138].

HNSCC cell lines resistant to irradiation showed global hypermethylation and differential methylation and expression of up to 84 genes relative to sensitive strains [278]. Among these, *CCND2*, a cell cycle controller, was hypermethylated in the promoter region [278]. *DAPK* promoter hypermethylation was detected in cell lines resistant to EGFR inhibitors, one of the few target therapies approved by the FDA to treat HNSCC patients [279,280]. Interestingly, *DAPK* knockout itself is able to induce resistance against these drugs [281]. In taxol-resistant nasopharyngeal carcinoma cell lines, global hypermethylation and 48 differentially methylated genes (30 hypermethylated and 18 hypomethylated) were observed, and treatment with demethylating agents could resensitize cells to chemotherapy [282]. Treatment with demethylating agents also sensitizes HNSCC cell lines to cisplatin [283].

Dozens of miRNAs were also found differentially expressed in HNSCC cell lines resistant to therapy [284,285]. Overexpression of let-7c, let-7d, let-7e, let-7g, and miR-20b [286], and downregulation of miR-200b, miR-15b [287], and miR-181a [288] were associated with platinum resistance.

More recently, FDA also approved immunotherapy based on PD1 and PD-L1 immune checkpoint blockade (pembrolizumab and nivolumab) for treatment of platinum-refractory metastatic HNSCC [289]. *PD-L1* and *PD-L2* promoter methylation levels are correlated with their expression and associated with HPV infection in HNSCC. Therefore, their assessment might be a promising predictive biomarker for treatment [290]. Furthermore, *PD-1* hypermethylation was already associated with a worse prognosis in HNSCC [291].

Regarding prognosis, worse overall and/or disease-free survival in HNSCC were associated with promoter hypermethylation of *KL* [130], *HIN1* [131], *RASSF1A*/*RASSF2* [131], *MGMT* [95,119,132,133], *DAPK* [119], *MINT31* [114], *p16INK4A*, and *p14ARF* [134]. Interestingly, *PAX5* and *PAX1* promoter methylation is associated with poorer overall survival in African-American patients compared to nonwhite Latinos [135].

A higher miR-21 expression or lower miR-375 expression was associated with a worse prognosis in LSCC [122]. In OPSCC, a panel with miR-142-3p, miR-31, miR-146a, miR-26b, miR-24, and miR-193b can predict survival regardless of HPV status or other clinical characteristics [292]. MiRNA signatures to predict overall survival (miR-107, miR-151, miR-492), disease-free survival (miR-107, miR-182, miR-20b, miR-151, miR-361), and distant relapse-free survival (miR324-5p, miR-492, miR-151, miR-361, miR-152) among OPSCC patients were described in a different study from the same year [293].

Epigenetic biomarkers in liquid biopsies also show promising results in predicting patient prognosis. *TIMP3* promoter hypermethylation detected in HNSCC patients’ saliva seems to be an independent marker of recurrence-free survival [136]. Higher *SEPT9* and *SHOX2* methylation levels in cell-free circulating DNA in the plasma of HNSCC patients were associated with worse overall survival and increased the chances of death approximately 5.2 and 2.3-fold, respectively [137]. When detected in the plasma of HNSCC patients, the high expression of miR-186-5p and miR-374b-5p before treatment and miR-142-3p after treatment were associated with reduced locoregional control. The expression of miR-142-3p was also associated with reduced disease-free progression [139].

### 4.4. The Epilogue of Epigenetic Biomarkers in HNSCC

The acquisition of epigenetic alterations due to risk factor exposure and transformation in a plastic but stable fashion, and their independence of genetic changes frequency present good potential for epigenetic biomarker development. These changes can be detected early during tumor development, enabling screening programs and increasing survival rates. Furthermore, the same marker has the potential to be used to predict more than one endpoint, such as diagnosis and recurrence, decreasing development time and cost. However, the establishment of the biomarker must be cautious, reproducible, and apply primarily sensitive methodologies.

In the case of DNA methylation, most studies cited here applied MSP or quantitative-MSP (qMSP) for biomarker identification, either directly or after selecting targets from microarray data. Although these techniques are quite user-friendly and generally available, they are subject to amplification biases and usually interrogate single CpG sites, which may impair the interpretation of results. Additionally, MSP does not provide a quantitative measure of methylation levels. This is especially relevant when we consider that pools of cells or of circulating DNA (in liquid biopsies) are usually analyzed. Since loci methylation will vary according to cell type and exposures, DNA methylation levels should be considered as a continuous and not a qualitative variable. Thus, bisulfite pyrosequencing or bisulfite amplicon–NGS could be more suitable for identifying potential biomarkers. Both techniques provide quantitative measures of DNA methylation levels and interrogate a higher number of CpG sites, but have a higher cost and require equipment that is not available in every research or clinical laboratory. Finally, all these techniques require a prior conversion of unmethylated cytosines into uracil by DNA treatment with sodium bisulfite. Despite its widespread use, this treatment might be an additional challenge for the analysis of poor-quality DNA. When these molecules are extracted from formalin-fixed paraffin-embedded (FFPE) tissues or from liquid biopsies, they are usually highly degraded, and the treatment with sodium bisulfite induces further degradation. However, the alternatives, such as the use of methylation-sensitive restriction enzymes or CpG methylation immunoprecipitation suffer from lower sensitivity. Based on these considerations, the ideal combination of techniques to evaluate DNA methylation alterations as biomarkers will depend on sample type, difference range, resources available, and biomarker sensitivity, among other factors. Nevertheless, after the establishment of methylation and discrimination levels, the construction of low-cost and more qualitative tests (such as qMSP) could be valid to decrease time and cost [294].

Although generally less explored than DNA methylation, miRNA-based biomarker development in HNSCC seems more advanced in terms of methodology. While many candidates are identified by microarray, a semiquantitative technique, their validation and assessment in other cohorts and sample sources are usually accomplished by reverse transcription followed by quantitative PCR (RT-qPCR). This is a widespread quantitative approach already applied in the clinics, as recently observed in COVID-19 diagnosis. The major challenge in the development of these biomarkers, however, is probably the definition of cutoffs for distinguishing groups, although this can be overcome by reproducibility studies, necessary for the development of any biomarker.

It is important to note that some genetic factors can affect the performance of the epigenetic biomarker. For example, *MTHFR* polymorphisms can affect *p16INK4A* and *MGMT* methylation levels in the oral cavity [295,296]. Even unexpected factors, such as pregnancy with intrauterine growth restriction, can cause aberrant methylation, such as *RASSF1A* hypermethylation [297], also observed in HNSCC [298]. In this context, the development of panels can be more sensitive and specific, avoiding interference from different factors and the generation of false negatives. The panel can be applied in minimally invasive approaches such as body fluids, where epigenetic changes are highly stable, or even in tumor biopsies, when available [299,300].

Until now, most of the proposed biomarkers are based on promoter hypermethylation of tumor suppressor genes, the mechanism generally involved in their silencing. Thus, the methylation status of these genes may also be indicative of the use of demethylating agents in the treatment, aiming for their reactivation. This bias towards promoters highlights the need for further studies characterizing the methylation profile in other genomic regions, such as intergenic regions, enhancers, and gene bodies.

Although less explored than DNA methylation and miRNAs, histone modifications have also been reported in HNSCC and have been associated with patients’ outcomes and cancer phenotypes. This is another potential field for the development of biomarkers, not addressed in this review (for further information, please refer to [285,301,302,303,304]), and should be further explored. In OSCC, H3K9ac reduction was associated with the activation of epithelial–mesenchyme transition, cell proliferation, and poor prognosis [305]; also in OSCC, the activation of Wnt/β-catenin signaling and the consequent induction of cell proliferation and invasion were associated with increased H3K27ac through the activation of *PLAC2*, a long-noncoding RNA [306]. Finally, high H3K27me3 and low H3K4ac levels were associated with poor prognosis [307], while reduced H3K4me3 and increased H3K4me2 levels were observed in OSCC relative to normal tissues [308], suggesting that histone methylation could also be an OSCC biomarker.

As mentioned previously, most studies considered all HNSCC sites as a single entity or focused on the oral cavity, probably due to the easier access and higher tumor occurrence. This might be an additional challenge because of differences in carcinogenesis mechanisms behind each specific subsite. Thus, the epilogue of HNSCC biomarkers is still blurry, and epigenetic changes can be an eye drop.

## 5. Conclusions

As highlighted in this review, epigenetic alterations are a common feature of HNSCC, and affect all steps in their natural history. In this context, intratumor heterogeneity, a major cause of cancer treatment failure and relapse, cannot be addressed only genetically. Epigenetics confers different phenotypes to subclones and, due to its reversibility, emerges as a potential therapeutic target. Therefore, to improve clinical management, it is of utmost importance that all professionals involved in patient care come to know this somewhat unexplored field.

Thus far, most studies have focused on the methylation of promoters of tumor suppressor genes. Although this approach has already identified putative carcinogenic pathways and biomarkers, much is yet to be learned from epigenetic alterations in HNSCC. Not only the methylation of other genes and other genomic regions, but also other players, including noncoding RNAs and chromatin remodelers, should be explored. For example, global hypomethylation, associated with poor prognosis in other cancer types [189,309,310,311], seems to differ according to HPV status and could help explain the prognosis differences observed. Furthermore, mutations in the histone methyltransferases *NSD1* and *NSD2* define a group of good prognosis among LSCC patients, but not in other HNSCC subtypes [312]. These data highlight the importance of painting a more detailed canvas of epigenetic landscape in HNSCC. The integration of different epigenetic mechanisms and the comprehension of both inter- and intratumoral heterogeneity may help achieve more personalized HNSCC patient care.

## Figures and Tables

**Figure 1 cancers-13-05630-f001:**
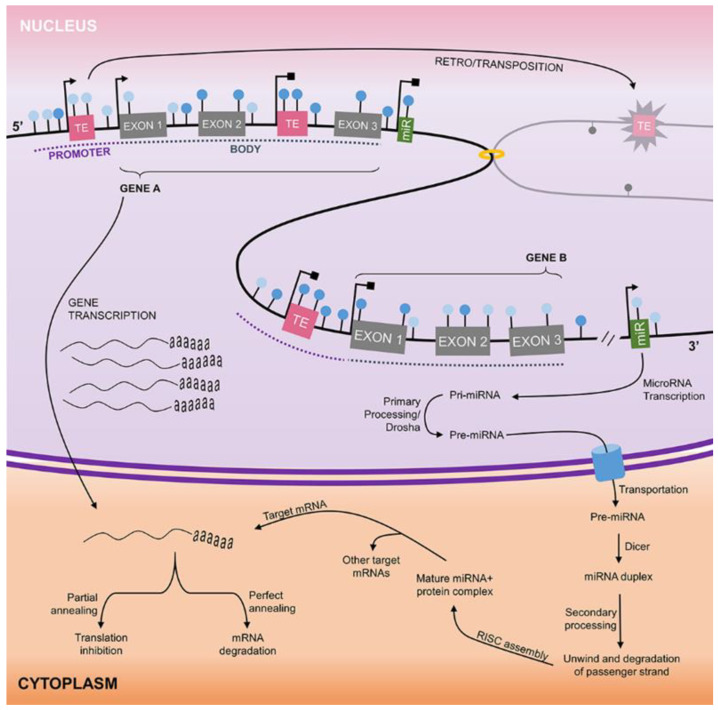
Overview of gene regulation by DNA methylation and miRNA activity. Genomic DNA, found in the nucleus of eukaryotic cells, contains millions of CpG sites (lollipops) that can be either methylated (dark blue) or demethylated (light blue). These CpG sites are located in gene promoters and bodies, miRNAs, and intergenic regions, as well as in the vicinity of transposable elements (TE). When a promoter region is demethylated (Gene A), gene transcription may take place, leading to mRNA synthesis. However, when the promoter is hypermethylated (Gene B), transcription is inhibited. TEs transcription goes through the same regulatory mechanism. The transcription of some of these elements leads to the production of the transposition machinery, which enables their retrotransposition to a new genomic region, in a “copy–paste” fashion. DNA methylation similarly regulates the expression of miRNAs. Once transcribed, the primary miRNA (Pri-miRNA) is processed still in the nucleus by the RNAse Drosha, forming the precursor miRNA (Pre-miRNA). The Pre-miRNA is exported to the cytoplasm, where it is further processed by the RNAse Dicer, forming the miRNA duplex. After the miRNA duplex unwinds and the degradation of the passenger strand takes place, the lead strand (mature miRNA) assembles to the RNA-induced silencing complex (RISC) and, by nucleotide complementarity, identifies their target mRNAs. A single miRNA may have several mRNA targets and will lead to their degradation (perfect annealing) or translation inhibition (partial annealing).

**Figure 2 cancers-13-05630-f002:**
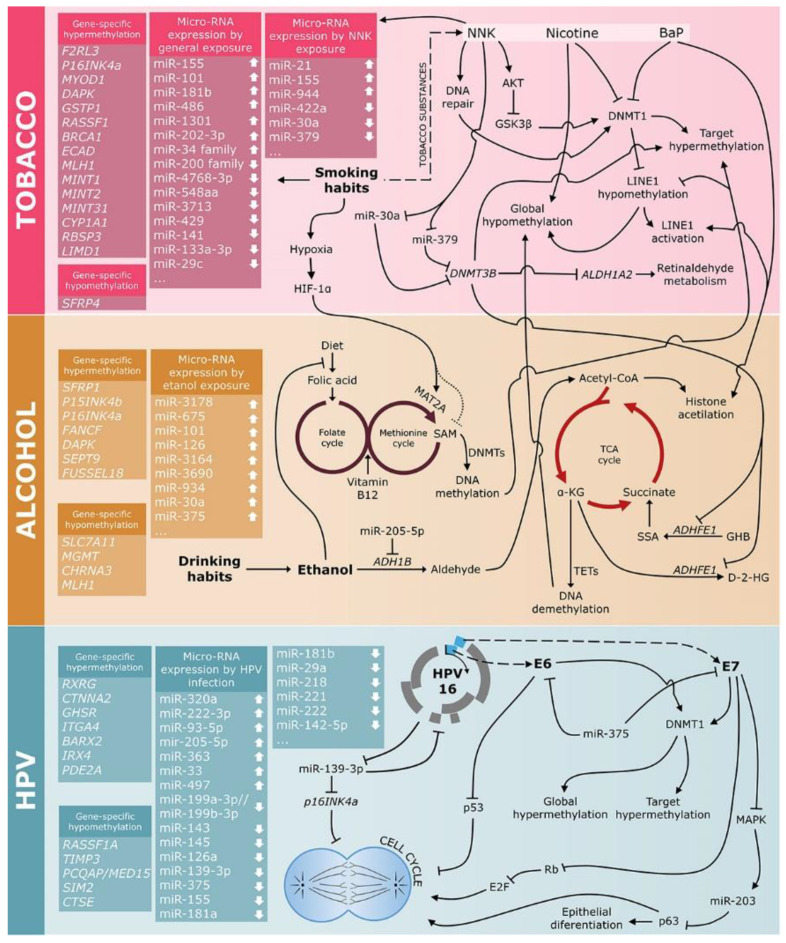
Risk factors associated with HNSCC development affect DNA methylation and miRNA expression, showing different mechanisms by which tobacco components, alcohol, and HPV infection may lead to aberrant DNA methylation profiles and aberrant miRNA expression. NNK, 4-(methylnitro-samino)-1-(3-pyridyl)-1-butanone; BaP, benzo(a)pyrene; TCA Cycle, tricarboxylic acid cycle.

**Figure 3 cancers-13-05630-f003:**
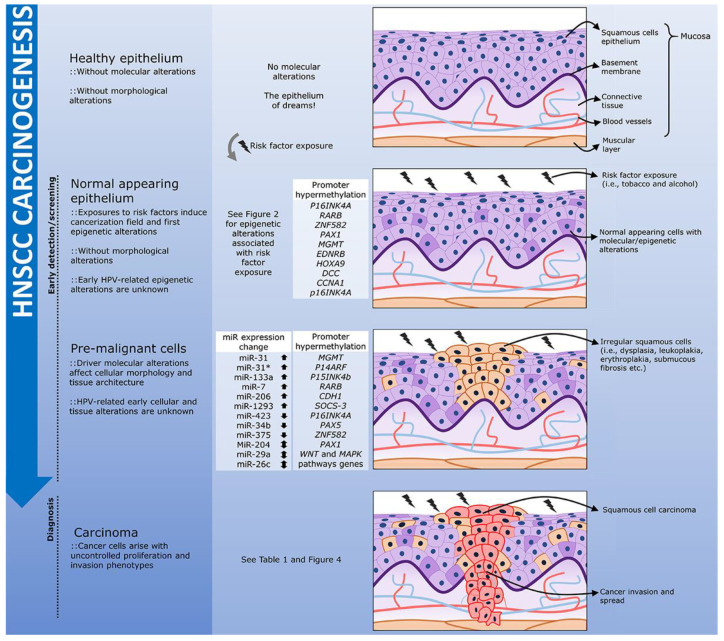
The natural history of HNSCC development through an epigenetic perspective. The exposure of the head and neck healthy epithelium to risk factors, such as tobacco and alcohol, may lead to the accumulation of DNA methylation alterations and miRNA aberrant expression even before the first morphological changes are observed. This characterizes a molecular cancerization field that may favor the development of pre-malignant lesions. In head and neck, the presence of leukoplakias and erythroplakias, frequently associated with dysplasia, increases the risk of tumor development, and these lesions are characterized by specific epigenetic alterations. The epigenetic aberrations observed in both the cancerization field and in the pre-malignant lesions, have the potential to be used as early markers of HNSCC carcinogenesis in screening programs. Finally, tumor onset and progression may be followed by further aberrant DNA methylation and miRNA expression. HPV-positive HNSCC epigenetic natural history is largely unknown.

**Figure 4 cancers-13-05630-f004:**
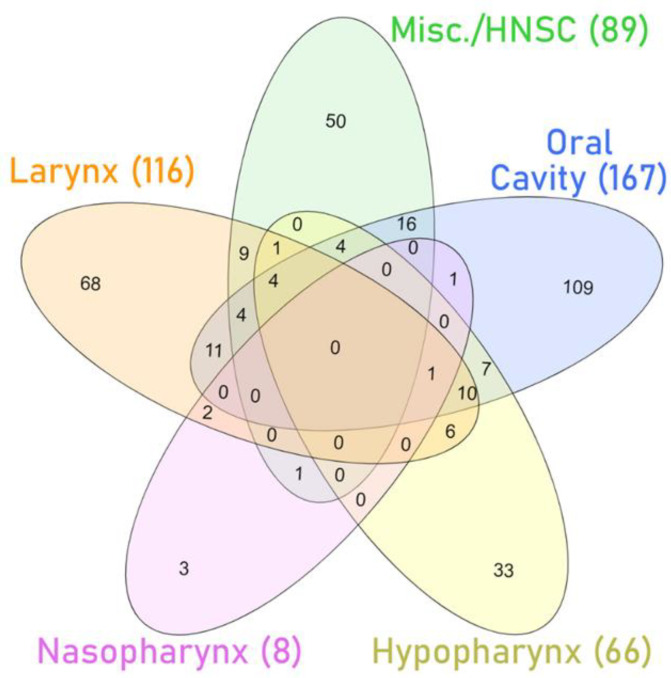
Venn diagram with the number of microRNAs differentially expressed (upregulated and downregulated) between tumors and nontumor tissues in several studies with specific subsets of HNSCC (larynx [241,242,243,244], nasopharynx [245], hypopharynx [246,247], and oral cavity [248,249,250,251]) or aggregating several sites in the analysis (Misc./HNSCC) [252,253,254]. The list of differentially expressed microRNAs can be found in Appendix A.

**Table 1 cancers-13-05630-t001:** Epigenetic biomarkers in head and neck squamous cell carcinoma.

Gene/Genomic Element	HNSCC Site	Sample Type	Genomic Region	Molecular Alteration	Ref.
Risk factor exposure: TOBACCO					
*p16INK4a* and *MYOD1*	HNSCC	Tissue	Promoter	Hypermethylation	[61]
*SFRP4*	HNSCC	Tissue	Promoter	Hypermethylation	[62]
*CYP1A1*	HNSCC	Tissue	Promoter	Hypermethylation	[63]
*RBSP3, LIMD1* and *CDC25A*	HNSCC	Tissue	Promoter	Hypermethylation	[64]
miR-200 family	OSCC	Tissue	N.A.	Downregulation	[65]
miR-155	OSCC	Tissue	N.A.	Upregulation	[66]
miR-29c	LSCC	Tissue	N.A.	Downregulation	[67]
miR-202-3p	LSCC	Tissue	N.A.	Upregulation	[68]
miR- 4768-3p, miR-548aa and miR-3713	LSCC	Tissue	N.A.	Downregulation	[68]
miR-34 family	HNSCC	Tissue	N.A.	Upregulation	[69]
mir-133a-3p	OPSCC	Tissue	N.A.	Downregulation	[70]
Risk factor exposure: ALCOHOL					
*p15INK4b*	HNSCC	Oral rinse	Promoter	Hypermethylation	[71]
*FUSSEL18* and *SEPT9*	HNSCC	Tissue	Promoter	Hypermethylation	[72]
*DAPK*	HNSCC	Tissue	Promoter	Hypermethylation	[73]
*SFRP1*	HNSCC	Tissue	Promoter	Hypermethylation	[62]
*p16INK4a, RASSF1A* and *FANCF*	HNSCC	Tissue	Promoter	Hypermethylation	[61,74]
*MGMT* and *CHRNA3*	HNSCC	Tissue	Promoter	Hypomethylation	[61]
*MLH1*	LSCC	Tissue	Promoter	Hypomethylation	[75]
miR-30a, miR-934, miR-3178, miR-675, miR-101, miR-126, miR-3164 and miR-3690	HNSCC	Tissue	N.A.	Upregulation	[48]
miR-375	HNSCC	Tissue	N.A.	Upregulation	[76]
Risk factor exposure: HPV					
*p16INK4a*	OSCC, OPSCC and HNSCC	Tissue and Saliva	Promoter	Hypermethylation	[77,78]
				Hypomethylation	[79,80]
*RASSF1A*, *TIMP3* and *PCQAP/MED15*	HNSCC	Saliva	Promoter	Hypomethylation	[79]
*RXRG*, *CTNNA2*, *GHSR* and *ITGA4*	OPSCC	Tissue	Promoter	Hypermethylation	[80]
Global methylation/Repetitive Element	HNSCC and OPSCC	Tissue	LINE1	Hypermethylation	[20]
miR-99a-3p and miR-4746-5p	HNSCC	Tissue	N.A.	Upregulation	[81]
miR-411-5p	HNSCC	Tissue	N.A.	Downregulation	[81]
miR-320a, miR-222-3p, and miR-93-5p	OPSCC	Tissue	N.A.	Upregulation	[82]
miR-199a-3p//miR-199b-3p, miR-143, miR-145, and miR-126a	OPSCC	Tissue	N.A.	Downregulation	[82]
Cancerization field biomarkers—alterations in premalignant lesions					
*SOCS-3*	HNSCC and dysplasias	Tissue	Promoter	Hypermethylation	[83]
*p16INK4a* and *MGMT*	OSCC and individuals at risk	Oral rinse and Blood	Promoter	Hypermethylation	[84,85]
*RARB*	OSCC precursor lesions	Tissue	Promoter	Hypermethylation	[86]
*ZNF582* and *PAX1*	OSCC cavity	Tissue	Promoter	Hypermethylation	[87]
*ZAP70* and *GP1BB*	Oral squamous intraepithelial lesions	Oral brushing	Exon 3/Exon 1	Hypermethylation/Hypomethylation	[88]
*miR-204*, *miR-31*, *miR-31**, *miR-133a*, *miR-7*, *miR-206*, and *miR-1293*	OSCC precursor lesions	Tissue	N.A.	Upregulation	[89]
Diagnostic biomarkers					
*ABO*	OSCC	Tissue	Promoter	Hypermethylation	[90]
*APC*	OSCC	Tissue	Promoter	Hypermethylation	[91]
*C/EBPα*	HNSCC	Tissue	Promoter	Hypermethylation	[92]
*CDH1*	OSCC	Tissue	Promoter	Hypermethylation	[91,93,94,95,96]
	HNSCC	Tissue	Promoter	Hypermethylation	[97]
	LSCC	Tissue	Promoter	Hypermethylation	[98]
	Hypopharynx	Tissue	Promoter	Hypermethylation	[98]
*CDKN2A*	OSCC	Tissue	Promoter	Hypermethylation	[91,93,94,95,96,99,100,101,102]
	HNSCC	Tissue	Promoter	Hypermethylation	[97,103]
	HNSCC	Saliva	Promoter	Hypermethylation	[102,104]
	LSCC	Tissue	Promoter	Hypermethylation	[75,98,105]
	Hypopharynx	Tissue	Promoter	Hypermethylation	[98]
*CHD5*	LSCC	Tissue	Promoter	Hypermethylation	[106]
*CYGB*	OSCC	Tissue	Promoter	Hypermethylation	[100]
*CYCA1*	OSCC	Tissue	Promoter	Hypermethylation	[93,96]
*DAPK*	OSCC	Tissue	Promoter	Hypermethylation	[73,91,94,99,101,107]
	OSCC	Blood	Promoter	Hypermethylation	[107]
	HNSCC	Tissue	Promoter	Hypermethylation	[73,97,102]
	HNSCC	Saliva	Promoter	Hypermethylation	[102,104]
	LSCC	Tissue	Promoter	Hypermethylation	[75,98,108]
	Hypopharynx	Tissue	Promoter	Hypermethylation	[98]
*DKK3*	OSCC	Tissue	Promoter	Hypermethylation	[109]
Global methylation/Repetitive Element	OSCC	Blood	LRE1/LINE1	Hypomethylation	[110]
	OSCC	Oral rinse	Alu	Hypomethylation	[111]
*HOXA9*	OSCC	Tissue and Saliva	Promoter	Hypermethylation	[112]
*MGMT*	OSCC	Tissue	Promoter	Hypermethylation	[91,94,95,96,99,101]
	HNSCC	Tissue	Promoter	Hypermethylation	[103,113]
	HNSCC	Saliva	Promoter	Hypermethylation	[102]
	LSCC	Tissue	Promoter	Hypermethylation	[75,98]
	Hypopharynx	Tissue	Promoter	Hypermethylation	[98]
*MLH1*	LSCC	Tissue	Promoter	Hypermethylation	[75,95]
	OSCC	Tissue	Promoter	Hypermethylation	
*MINT1*	OSCC	Tissue	Promoter	Hypermethylation	[114]
*NID2*	OSCC	Tissue and Saliva	Promoter	Hypermethylation	[112]
*p14*	HNSCC	Tissue	Promoter	Hypermethylation	[103,115]
*p15*	OSCC	Tissue	Promoter	Hypermethylation	[95,116]
*PCQAP/MED15*	OPSCC	Saliva	Promoter	Hypermethylation	[117]
	OSCC	Saliva	Promoter	Hypermethylation	[117]
*RARβ*	OSCC	Tissue	Promoter	Hypermethylation	[93,96]
	HNSCC	Tissue	Promoter	Hypermethylation	[103]
*RARβ2*	Salivary Gland Carcinomas	Tissue	Promoter	Hypermethylation	[118]
*RASSF1A*	OSCC	Tissue	Promoter	Hypermethylation	[119]
	Salivary Gland Carcinomas	Tissue	Promoter	Hypermethylation	[118]
	HNSCC	Tissue	Promoter	Hypermethylation	[97]
	HNSCC	Saliva	Promoter	Hypermethylation	[104]
	OPSCC	Saliva	Promoter	Hypermethylation	[117]
*RUNX3*	OSCC	Tissue	Promoter	Hypermethylation	[94]
*SFRP1*	OSCC	Tissue	Promoter	Hypomethylation	[109]
*SFRP2*	OSCC	Tissue	Promoter	Hypermethylation	[109]
*SFRP4*	OSCC	Tissue	Promoter	Hypermethylation	[109]
*SFRP5*	OSCC	Tissue	Promoter	Hypermethylation	[109]
*SOCS-3*	HNSCC	Tissue	Promoter	Hypermethylation	[83]
*SSTR2*	LSCC	Tissue	Promoter	Hypermethylation	[120]
*WIF1*	OSCC	Tissue	Promoter	Hypermethylation	[109,119]
*WRN*	OSCC	Tissue	Promoter	Hypermethylation	[91]
*ZNF14*	HNSCC	Tissue and Saliva	Promoter	Hypermethylation	[121]
*ZNF160*	HNSCC	Tissue and Saliva	Promoter	Hypermethylation	[121]
*ZNF420*	HNSCC	Tissue and Saliva	Promoter	Hypermethylation	[121]
miR-375	LSCC	Tissue	N.A.	Downregulation	[122]
miR-21	LSCC	Tissue	N.A.	Upregulation	[122]
miR-204 and mir-144	OSCC	Tissue and plasma	N.A.	Upregulation	[123]
miR-193b-5p and miR-370-3p	OSCC	Tissue and plasma	N.A.	Downregulation	[123]
miR-657	LSCC	Tissue	N.A.	Upregulation	[124]
miR-1287	LSCC	Tissue	N.A.	Downregulation	[124]
Relapse biomarkers					
*EDNRB* and *HOXA9*	HNSCC surgical margins	Tissue	Promoter	Hypermethylation	[125]
*DCC*, *CCNA1* and *p16INK4A*	HNSCC surgical margins	Tissue	Promoter	Hypermethylation	[126]
*PAX5*	HNSCC surgical margins	Tissue	Promoter	Hypermethylation	[127]
*CDKN2A*	LSCC	Tissue	Nonpromoter	Hypomethylation	[128]
miR-375	OSCC precursor lesions	Tissue	N.A.	Downregulation	[129]
Biomarkers of poor prognosis					
*KL*	HNSCC	Tissue	Promoter	Hypermethylation	[130]
*HIN1*, *RASSF1A* and *RASSF2*	OSCC	Tissue	Promoter	Hypermethylation	[131]
*DAPK*	OSCC	Tissue	Promoter	Hypermethylation	[119]
*MGMT*	OSCC	Tissue	Promoter	Hypermethylation	[95,119,132,133]
*MINT31*	OSCC	Tissue	Promoter	Hypermethylation	[114]
*p16INK4A* and *p14ARF*	OSCC	Tissue	Promoter	Hypermethylation	[134]
*PAX5* and *PAX1*	HNSCC	Tissue	Promoter	Hypermethylation	[135]
*TIMP3*	HNSCC	Saliva	Promoter	Hypermethylation	[136]
*SEPT9* and *SHOX2*	HNSCC	Plasma	Promoter	Hypermethylation	[137]
*EPHX3*	OSCC	Pre-operative oral brushing	Exon 1	Hypomethylation	[138]
*ITGA4* and *MIR193A*	OSCC	Pre-operative oral brushing	Exon 2/Promoter	Hypermethylation	[138]
miR-21	LSCC	Tissue	N.A.	Upregulation	[122]
miR-375	LSCC	Tissue	N.A.	Downregulation	[122]
miR-186-5p and miR-374b-5p	HNSCC	Plasma—before treatment	N.A.	Upregulation	[139]
miR-142-3p	HNSCC	Plasma—after treatment	N.A.	Upregulation	[139]

HNSCC, head and neck squamous cell carcinoma; LSCC, laryngeal squamous cell carcinoma; N.A., not applicable; OPSCC, oropharyngeal squamous cell carcinoma; OSCC, oral squamous cell carcinoma; Ref., references.

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
