# Peer review of "Head and Neck Cancers Are Not Alike When Tarred with the Same Brush: An Epigenetic Perspective from the Cancerization Field to Prognosis"

_cancers, 2021, doi:10.3390/cancers13225630_

Round 1

Reviewer 1 Report

The manuscript entitled "Head and Neck cancers are not alike when tarred with the same brush: an epigenetic perspective from the cancerization field to prognosis" describes a very extensive literature review of the state-of-the-art research in the field of HNSCC. It covers mostly DNA methylation and micro-RNAs expression as the main 'routes' of epigenetic modifications that shape HNSCC natural history.

It is a very well written detailed review manuscript. Figures are comprehensive and detailed. More specific comments below:

Sections 1 & 2 are probably a bit long given the length of this review; Lines 75-82 of Section 1 sound repetitive or maybe could be placed further in another section. Please consider shortening these sections for ease of reading.

Please consider reorganizing subsections 4.1, 4.2 and 4.3 as 'tobacco', followed by 'alcohol' and 'HPV' as represented in Figure 2.

Little is referenced about techniques and how that could influence study comparisons; similarly, little is attributed to next-generation sequencing, one of the leading techniques nowadays. Though it is understandable that this field of research it is still in its infancy, affordability and depth of sequencing/data/information could be key to discovery. NGS has the potential to shed light to many of the questions raised here and it provides more data that the most commonly used Q-MSP.

This review aims to provide detail on what is known so far for HNSCC and explore/enumerate some biomarkers; hence, discussing strengths and/or limitations of available techniques may be an important statement. Please consider shorten some of the existing sections and include some relevant technical discussion.

Author Response

The manuscript entitled "Head and Neck cancers are not alike when tarred with the same brush: an epigenetic perspective from the cancerization field to prognosis" describes a very extensive literature review of the state-of-the-art research in the field of HNSCC. It covers mostly DNA methylation and micro-RNAs expression as the main 'routes' of epigenetic modifications that shape HNSCC natural history.

It is a very well written detailed review manuscript. Figures are comprehensive and detailed. More specific comments below:

  1. Sections 1 & 2 are probably a bit long given the length of this review; Lines 75-82 of Section 1 sound repetitive or maybe could be placed further in another section. Please consider shortening these sections for ease of reading.

ANSWER: We thank the reviewer for the comments and suggestions. We agree that sections 1 & 2 were too long, so we reduced and merged them in the revised manuscript.

  1. Please consider reorganizing subsections 4.1, 4.2 and 4.3 as 'tobacco', followed by 'alcohol' and 'HPV' as represented in Figure 2.

ANSWER: Subsections were reordered as suggested.

  1. Little is referenced about techniques and how that could influence study comparisons; similarly, little is attributed to next-generation sequencing, one of the leading techniques nowadays. Though it is understandable that this field of research it is still in its infancy, affordability and depth of sequencing/data/information could be key to discovery. NGS has the potential to shed light to many of the questions raised here and it provides more data that the most commonly used Q-MSP.

ANSWER: We thank the reviewer for raising this point and we added two studies which evaluated DNA methylation biomarkers by NGS in oral cancer (lines 588-597, 736-740). To further highlight the potential of this technique in biomarkers assessment, a paragraph on technical advantages/limitations was added to the manuscript and NGS was included in this discussion (lines 794-827).

  1. This review aims to provide detail on what is known so far for HNSCC and explore/enumerate some biomarkers; hence, discussing strengths and/or limitations of available techniques may be an important statement. Please consider shorten some of the existing sections and include some relevant technical discussion.

ANSWER: Following the reviewer’s suggestion, we included the technical discussion in the revised manuscript (lines 794-827).

Reviewer 2 Report

Dear Authors,

The article: 'Head and Neck cancers are not alike when tarred with the same brush: an epigenetic perspective from the cancerization field to prognosis' was to summarize knowledge about HNC and epigenenic prognosis'.

English language and style are fine.

Punctuation mistakes should be corrected. 

The article is well planned and prepared. The aim of the work should be better formulated. It contains a decent summary of the analyzed topic.

To sum up, article can be accepted after minor revision.

Author Response

Dear Authors,

The article: 'Head and Neck cancers are not alike when tarred with the same brush: an epigenetic perspective from the cancerization field to prognosis' was to summarize knowledge about HNC and epigenenic prognosis'.

English language and style are fine.

  1. Punctuation mistakes should be corrected.

ANSWER: We thank the reviewer for the comments and suggestions. Punctuation was revised throughout the text.

  1. The article is well planned and prepared. The aim of the work should be better formulated. It contains a decent summary of the analyzed topic.

ANSWER: As suggested, the aim of the work was reformulated to highlight the information brought in the manuscript (lines 111-115).

To sum up, article can be accepted after minor revision.

Reviewer 3 Report

The review article “Head and Neck cancers are not alike when tarred with the same brush: an epigenetic perspective from the cancerization field to prognosis” by Camuzi et al. summarizes and discusses a wide range of literature (over 300 publications) about epigenetic alterations, including DNA methylation and miRNAs, observed in HNSCC. The Authors provide up-to-date and comprehensive data on changes in DNA methylation (mostly promoter methylation) and miRNAs expression associated with HNSCC development. They depicted those epigenetic signatures of head and neck cancers as pivotal targets for therapeutic strategies, as well as potential candidates for biomarkers of diagnosis and prognosis for HNSCC, taking into account different risk factors (tobacco, alcohol and HPV infection). The review is novel and includes a clear and comprehensive table, as well as informative figures. The Authors provide a lot of information supported by abundant literature. The data are well organized and clear.

The epigenetic landscape consists of molecular events, including not only DNA methylation and non-coding RNA-related mechanisms (miRNAs), but also chromatin remodeling. Although, the Authors mentioned the following argument: “Much progress has been made in evaluating alterations of DNA methylation and non-coding RNAs in HNSCC. Due to analytical advantages compared with chromatin remodeling, these epigenetic marks will be the focus of our discussion here.”, it would be beneficial to briefly summarize the histone marks observed in HNSCC and their potential as biomarkers of HNSCC.

Author Response

The review article “Head and Neck cancers are not alike when tarred with the same brush: an epigenetic perspective from the cancerization field to prognosis” by Camuzi et al. summarizes and discusses a wide range of literature (over 300 publications) about epigenetic alterations, including DNA methylation and miRNAs, observed in HNSCC. The Authors provide up-to-date and comprehensive data on changes in DNA methylation (mostly promoter methylation) and miRNAs expression associated with HNSCC development. They depicted those epigenetic signatures of head and neck cancers as pivotal targets for therapeutic strategies, as well as potential candidates for biomarkers of diagnosis and prognosis for HNSCC, taking into account different risk factors (tobacco, alcohol and HPV infection). The review is novel and includes a clear and comprehensive table, as well as informative figures. The Authors provide a lot of information supported by abundant literature. The data are well organized and clear.

  1. The epigenetic landscape consists of molecular events, including not only DNA methylation and non-coding RNA-related mechanisms (miRNAs), but also chromatin remodeling. Although, the Authors mentioned the following argument: “Much progress has been made in evaluating alterations of DNA methylation and non-coding RNAs in HNSCC. Due to analytical advantages compared with chromatin remodeling, these epigenetic marks will be the focus of our discussion here.”, it would be beneficial to briefly summarize the histone marks observed in HNSCC and their potential as biomarkers of HNSCC.

ANSWER: We thank the reviewer for the comments and suggestions. Following the recommendations, we added a paragraph on the histone modifications observed in HNSCC and their relevance as biomarkers (lines 843-854).